# Prognostic Factors Related to Clinical Response in 210 Knees Treated by Platelet-Rich Plasma for Osteoarthritis

**DOI:** 10.3390/diagnostics13040760

**Published:** 2023-02-17

**Authors:** Clément Chopin, Marion Geoffroy, Lukshe Kanagaratnam, Claire Dorilleau, Fiona Ecarnot, Renaud Siboni, Jean-Hugues Salmon

**Affiliations:** 1Department of Rheumatology, University of Reims Champagne-Ardenne (URCA), Reims University Hospital, 51100 Reims, France; 2Department of Rheumatology, Reims University Hospital, 51100 Reims, France; 3Department of Research and Public Health, Reims University Hospital, 45, Rue Cognacq-Jay, 51092 Reims, France; 4EA3920, Department of Cardiology, University of Franche-Comté, 25000 Besançon, France; 5Department of Orthopaedic Surgery, Reims University Hospitals, 51100 Reims, France

**Keywords:** osteoarthritis, knee, platelet-rich plasma, cartilage, frailty, musculoskeletal disorders

## Abstract

Many studies have shown the effectiveness of platelet-rich plasma (PRP) in the treatment of knee osteoarthritis. We aimed to determine the factors associated with good or poor response to PRP injections in knee osteoarthritis. This was a prospective observational study. Patients with knee osteoarthritis were recruited from a university hospital. PRP was injected twice at a one-month interval. Pain was assessed on a visual analog scale (VAS) and function was assessed using the Western Ontario and McMaster Universities Osteoarthritis Index (WOMAC). Radiographic stage was collected and defined according to the Kellgren–Lawrence classification. Patients were classified as responders if they met the OMERACT-OARSI criteria at 7 months. We included 210 knees. At 7 months, 43.8% were classified as responders. Total WOMAC and VAS were significantly improved between M0 and M7. Physical therapy and a heel–buttock distance >35 cm were the two criteria associated with poor response at M7 by multivariate analysis. Pain VAS at M7 appeared to be lower in patients with osteoarthritis for less than 24 months. No adverse effects were reported. PRP treatment in knee osteoarthritis appears to be well-tolerated and effective, even in patients who reacted poorly to hyaluronic acid. Response was not associated with radiographic stage.

## 1. Introduction

Knee osteoarthritis (KOA) is a frequent and debilitating degenerative condition that causes limitations in daily activities and functional disability [1]. In 2020, Cui et al. estimated the global prevalence of knee OA to be 16.0% (95% CI, 14.3–17.8%) in individuals aged 15 and over and 22.9% (95% CI, 19.8–26.1%) in individuals aged 40 and over [2]. Buckwalter et al. estimated that USD 3.4–13.2 billion is spent annually on job-related OA costs in the USA [3], while in France, the KHOALA cohort reported that the median costs induced by KOA could be as high as EUR 2 billion per year (IQR 0.7–4.3) [4].

The European League Against Rheumatism (EULAR) and Osteoarthritis Research Society International (OARSI) recommend using a combination of pharmacological and non-pharmacological approaches for the management of KOA [5,6,7]. Among the pharmacological approaches, intra-articular corticosteroids (IACSs) occupy a leading role [8]. However, their use is not devoid of risk, and their medium-to-long-term efficacy is mediocre. Hyaluronic acid is another therapeutic option and has better efficacy and tolerance than IACS. However, its efficacy for pain reduction in the short-term is lower [9], and no structural effect of hyaluronic acid has been demonstrated [10]. However, debate persists surrounding the safety of viscosupplementation, with a recent meta-analysis reporting a significantly higher risk of serious adverse events with this treatment compared to placebo [11]. Moreover, hyaluronic acid is no longer reimbursed in certain countries. 

Over the last few years, new therapies have been investigated for the treatment of KOA, including intra-articular (IA) platelet-rich plasma (PRP) [12]. PRP can be obtained quickly and simply by centrifugation of a blood sample. Numerous studies have investigated the efficacy of PRP in KOA, and meta-analyses have reported significant pain reduction and functional improvement with PRP [13,14,15,16] and a more prolonged effect compared to hyaluronic acid [15,17]. However, the protocols for PRP administration and the formulations used vary widely in the setting of KOA, rendering comparison across studies difficult. For this reason, professional societies have not yet recommended PRP for the management of KOA. A group of French-speaking experts recently issued a consensus statement with guidelines for PRP injections in KOA [18], recommending this approach as a second-line treatment after the failure of per os and/or non-pharmacological approaches. However, at present, it remains difficult to predict the response to PRP and therefore, to identify patients most likely to benefit from this treatment.

The aim of our study was therefore to investigate the associations between the baseline characteristics of KOA patients to identify factors that predict response to PRP injection.

## 2. Materials and Methods

### 2.1. Study Population and Outcomes

This was a single-center observational study conducted at the University Hospital of Reims using data routinely recorded in the medical files of patients treated for KOA at our outpatient clinic from 24 May 2019 to 12 February 2021. All patients had symptomatic KOA meeting the American College of Rheumatology (ACR) criteria [19], had failed usual first-line therapy, and were referred by their rheumatologists to our outpatient clinic. In this clinic, patients have a clinical interview and clinical evaluation by a rheumatologist, an interview with an orthopedic surgeon, a group evaluation by a dietician and physiotherapist. After this process, patients in our study then received the first injection of PRP under ultrasound guidance and a second injection one month later during an outpatient consultation. 

Prior to referral, patients had a knee x-ray to classify KOA according to the Kellgren–Lawrence classification (grade 0 to 4) [20] and to check for any static disorders. Patients were instructed to limit, and if possible, interrupt their intake of non-steroidal anti-inflammatory drugs (NSAIDs) in the 10 days before PRP injection. 

Patient data were collected using a checklist that included socio-demographic characteristics, waist circumference (to determine the presence or absence of metabolic syndrome), prior medical history, and Charlson comorbidity index [21]. We also noted the use of painkillers and non-pharmacological approaches (e.g., knee braces, physiotherapy, physical activity), the time since the onset of symptoms, the number of viscosupplementations previously received, and whether a corticosteroid injection had been performed in the 3 months preceding the procedure. The general beliefs dimension of the Beliefs about Medicines Questionnaire was also administered to assess patients’ beliefs relating to medication use [22].

Patients’ function was evaluated using two scores, namely the Western Ontario and McMaster Universities Osteoarthritis Index (WOMAC) total score [23] and the Knee injury and Osteoarthritis Outcome Score (KOOS-PS) [24], and pain was evaluated on a visual analog scale (VAS) (from 0 (no pain) to 10 (maximal pain)), at 0, 1, 4, and 7 months (M0, M1, M4, M7) after the first injection. Heel-to-buttock distance was measured at baseline and at 1 month, during the second injection.

The primary endpoint was responder or non-responder status, defined at 7 months. Patients were classified as responders if they met the OMERACT-OARSI criteria at M7 and as non-responders if not, or if a new injection or surgery had been performed during follow-up. OMERACT-OARSI response criteria were defined as follows [25]:-Improvement in pain or in function ≥ 50% and absolute change ≥ 20;-**Or** improvement in at least two of the following:Improvement in pain VAS score ≥ 20% and absolute change ≥ 10;Improvement in function ≥ 20% and absolute change ≥ 10;Improvement in patient’s global assessment ≥ 20% and absolute change ≥ 10.

Since the function subsection of the WOMAC was not available for all patients, we used the WOMAC total score to assess the function criterion. 

Secondary endpoints were: change in WOMAC total score, VAS pain score, and KOOS-PS score; achievement of patient-acceptable symptom state (PASS) (defined as 40 out of 100 for chronic pain in rheumatic diseases [26]) or a WOMAC total score ≤ 46. We also recorded Minimal Clinically Important Improvement (MCII) in pain, defined as a reduction in VAS pain score ≥ 15/100. 

Patient-reported assessment of treatment efficacy, patient-reported overall satisfaction with treatment, and the reduction in the use of painkillers were evaluated using a four-point Likert scale.

### 2.2. Preparation of PRP 

Platelet-rich plasma (PRP) formulations were prepared using the Arthrex ACP ^®^ system. A nurse drew a blood sample of 15 mL, then the sample was centrifuged according to the manufacturer’s instructions (5 min at 1500 rpm) to obtain 3 to 6 mL of PRP, ready for use with the double-syringe system [27]. According to Sundman et al. [28], this PRP had low white blood cell content (×0.13 compared to whole blood) and platelets were concentrated ×1.99 [29] to obtain pure PRP [30]. Compared with controls (white blood cell count, 8.73 × 10^9^/L), Arthrex ACP^®^ reduces the white blood cell count by almost eliminating the white blood cells to 1.3 × 10^9^/L. The ACP^®^ kit had negligible lymphocytes (0.7 × 10^9^/L) and neutrophil count (0.4 × 10^9^/L) [29]. Therefore, the PRP used is an LP-PRP which is recommended for KOA according to Milants et al. [31]. The PRP is then administered as an IA injection under echographic guidance at M0. A second injection is performed at M1 after preparation of PRP according to the same protocol. The choice to perform one round of centrifugation and to administer 2 injections at a 1-month interval was made based on the data described by Milants. If joint effusion occurred, needle aspiration was performed at the same time as PRP injection.

### 2.3. Statistical Analysis

Quantitative variables are described as mean ± standard deviation if normally distributed or as median and interquartiles (Quartile 1 (Q1), Quartile 3 (Q3)) if non-normally distributed. Discrete variables are described as a number and percentage and were compared between groups using the chi square or Fisher’s exact test. The Wilcoxon Mann–Whitney and Student’s t tests were used to compare VAS, WOMAC, and KOOS-PS scores, as appropriate, and the different follow-up timepoints. Paired tests were used where appropriate. All tests were two-sided, and a *p*-value < 0.05 was considered statistically significant. All variables with a *p*-value < 0.20 by univariate analysis were included in the multivariate analysis. A manual backward selection procedure was used to define the final model. Multivariate analysis was systematically adjusted for VAS and WOMAC. Results are presented as odds ratios (ORs) with 95% confidence intervals (CIs). All analyses were performed using SAS version 9.4 (SAS Institute Inc., Cary, NC, USA).

### 2.4. Ethical Considerations

The study was approved by the French national authority for data privacy (CNIL, Commission Nationale de l’informatique et des libertés).

## 3. Results

### 3.1. Study Population 

From among 219 eligible patients, 197 patients were included, totaling 210 knees treated during the study period. The flowchart of the study population is shown in Figure 1. The majority were women and the average age was 59.8 ± 10.8 years. The characteristics of the study population are shown in Table 1.

Internal femorotibial OA was the most common, and the majority had advanced disease (50.7% with Kellgren–Lawrence grade ≥3). Only 16.8% suffered joint effusion. Eighteen patients (9.1%) had associated chrondrocalcinosis, and less than half (42.4%) were viscosupplement-naive.

### 3.2. Efficacy and Patient Satisfaction

The primary outcome of OMERACT-OARSI response was achieved for 92 of the 210 knees at M7 (43.1%). The MCII for VAS pain was achieved in 38.1%, 47.9%, and 46.5% of patients at M1, M4, and M7.

The mean improvement in WOMAC score was 20.8 ± 19.6 at M4 and 19.5 ± 20.9 at M7 versus baseline. Changes in WOMAC, KOOS-PS, and VAS scores during follow-up are described in Table 2. All improvements were significant at all timepoints (M1, M4, and M7) compared to baseline, except for the physical function subscale of the WOMAC, which was significantly improved at M4 and M7 only. There was also a significant difference between M1 and M4 and between M1 and M7 for all variables. Conversely, no significant differences were observed between M4 and M7.

The heel-to-buttock distance was significantly improved at 1 month after the first injection, increasing from 27.3 ± 11.4 to 20.8 ± 8.0 (*p* < 0.0001). 

At 1 month, among 174 patients with self-reported treatment efficacy, 95 (55%) judged the treatment to be efficacious or very efficacious on a 4-point Likert scale, and 35% reported having improved by more than 50%. These rates increased to 69% and 57% at M4 and 70% and 59% at M7. At M1, 28% had a reduction of more than 50% in their use of painkillers, and this increased to 56% at M4 and 54% at M7. Only 25% and 27% were slightly or very dissatisfied at M4 and M7. 

### 3.3. Comparison of Responders vs. Non-Responders 

By univariate analysis, comparing responders to non-responders, only the heel-to-buttock distance and the use of physiotherapy were significantly different between groups (Table 3).

There was no difference between groups in terms of physical exercise or the use of a knee brace, history of viscosupplementation, comorbidities (e.g., hypertension, diabetes, dyslipidemia), or KOA phenotype (hereditary, due to inflammatory diseases, metabolic, trauma, or static disorders). The BMQ scores and the conviction of effectiveness did not differ between the two groups. In terms of radiographic stage, 54% (13/24) of Kellgren I, 41% (28/68) of Kellgren II, and 45% (39/82) of Kellgren III were good responders versus 34% of those with Kellgren–Lawrence grade IV (10/29), and the differences between groups were not significant. 

When comparing patients with a more recent (<24 months) onset of KOA symptoms, there were numerically more respondents (50.8% (32/63), compared to 40.8% (60/147) in those with symptoms onset ≥24 months), but the difference was not statistically significant (*p* = 0.22). There was no significant difference in the changes in VAS, WOMAC, or KOOS-PS scores according to the time since symptom onset. 

By multivariate analysis (Table 4), patients receiving physiotherapy (adjusted OR 0.46, 95%CI 0.24–0.90) and those with a heel-to-buttock distance >35 cm (adjusted OR 0.31, 95%CI 0.15–0.65) were less likely to be responders. 

Initial WOMAC and VAS pain scores were not significantly associated with the response to PRP injection.

### 3.4. Tolerance

In terms of tolerance, no episode of infection was observed in any patient during follow-up. A total of 24/35 knees (69%) had persisting joint effusion between M0 and M1, and the onset of effusion was observed in 50/150 (33%). In patients with effusion at M1, the VAS score was 48.6 ± 23.9 at M1 versus 52.0 ± 20.7 for those without effusion, *p* = 0.34. The change in VAS score was similar in both groups.

## 4. Discussion

In this study, we sought to investigate the association between baseline characteristics and response to PRP injection in patients with osteoarthritis of the knee. We did not observe any effect of age, duration of KOA, body mass index, radiographic severity, phenotype, or localization on response to PRP. Only a heel-to-buttock distance > 35 cm and ongoing physiotherapy were associated with poorer response to treatment in this study. 

There are several possible explanations for these findings. Firstly, the heel-to-buttock distance is a simple clinical metric reflecting the functional repercussions of KOA. This distance can be increased by the presence of joint effusion, as well as more advanced disease with a limited degree of flexion. Regarding the association between physiotherapy and poor response, it is likely that patients with more advanced disease receive physiotherapy and have greater functional disability. Perhaps excessive mobilization of the joint during physiotherapy after the injection may have mitigated the efficacy of PRP. In our injection protocol, as suggested by Eymard et al. [18], we recommended 48 h of rest after the injection, and some teams recommend an even longer period of recovery before resuming activity. 

Few studies to date have investigated the predictors of good or poor response to PRP injections in KOA. Certain criteria, such as age, overweight, or the severity of KOA have been reported to be related to the response to PRP treatment [18]. The results of different studies investigating PRP are summarized in Table 5. Studies in vitro showed variations in the growth factors contained in PRP according to patient characteristics. Evanson et al. [32] described increased levels of Epidermal Growth Factor (EGF), Hepatocyte Growth Factor (HGF), Insulin-like Growth Factor 1 (IGF-1), and Platelet-Derived Growth Factor-BB human (PDGF-BB) in women and elevated levels of EGF, IGF-1, PDGF-AB, PDGF-BB, and transforming growth factor-β (TGFβ-1) in those aged under 25 years. In another in vitro study, O’Donnell [33] reported a reduction in chondrogenic markers (decreases in the expression of Col2a1 and Sox-9 mRNA by 40% and 30% respectively) in PRP from older (62–85 years old) male patients with severe KOA, compared to younger, healthy subjects. These data could explain the differences in efficacy in favor of younger patients. In a study by Kon et al. [34] (*n* = 150), PRP was found to be more effective in younger (aged under 50 years) and active patients with early-to-moderate OA (Kellgren–Lawrence grade 0 to III). Similar findings were reported in an Italian study of 144 symptomatic patients affected by cartilage degenerative lesions and OA [35]. This same team reported in another study [36] that body weight was correlated with clinical outcome after a cycle of three weekly injections of autologous conditioned plasma, whereby higher body mass index was associated with poorer response and a trend towards a shorter duration of efficacy. Conversely, in our study, and in the study by Patel et al. [37], no relation between body mass index and outcome was observed. The impact of body weight therefore remains uncertain, and further studies are warranted to elucidate the possible role of weight on the efficacy of PRP. 

Regarding the degree of severity of KOA, typically assessed using the Kellgren–Lawrence classification [20], it seems logical that more severe forms of disease respond less well to PRP treatment than less advanced forms of disease. Such a relation has previously been described for hyaluronic acid [38] and seems also to hold for PRP, as shown in the study by Saita [12]. In their study performed among a large population (517 knees), including 42% with severe (Kellgren–Lawrence grade IV) disease, efficacy was lower among those with severe disease, compared to those with mild-to-moderate forms.
diagnostics-13-00760-t005_Table 5Table 5Factors associated with response to PRP by multivariable analysis.AuthorPRP Protocol*n*Control GroupMean BMIPrior HADisease Duration (Years) Median (Q1–Q3)KL IVPrimary OutcomeFactors Associated with Lack of EfficacyKon et al.,2011 [34]3 (/15 days)150Yes (HA)24.6 ± 3.2NRNC16%IKDC at 6 MAge Physical activityHigh KL gradeFilardo et al.,2012 [35]151No27 (21–39)67%4 (1.5–10)20%IKDCHigh BMI High KL gradePatel et al.,2013 [37]2 (/21 days)156Yes (NaCl)25.8 ± 3.3NRNRNRWOMAC3 M and 6 MAhlbäck classificationSaita et al.,2021 [39]3 (/28 days)517No25.0NRNR42%OMERACT-OARSI response at 6 M and 12 M KL IVAlessio-Mazzola etal., 2021 [40]4 (/7 days)118No25.9 ± 2.450%NR0%Failure (Surgeryor recurrence at M6) KL IIIHigh BMI Chopin et al., 20222 (/28 days)210No31.0 ± 6.558%3 (1–6)14%OMERACT-OARSIat 7 MPhysiotherapyHeel-to-buttock distance >35 cmPRP, platelet rich plasma; BMI, body mass index; HA, hyaluronic acid; Q, quartile; KL IV, Kellgren–Lawrence grade IV; NR, not reported; IKDC, International Knee Documentation Committee; M, months; WOMAC, Western Ontario and McMaster Universities Arthritis Index.

In our study, 58% of patients had failed viscosupplementation, which is comparable to the rate reported by Alessio-Mazzola et al. (50%) [40]. However, this rate is rarely reported in other studies, and the same is true for the duration of symptoms. Among 92 responders in our study, 52 had failed viscosupplementation (56.5%), which is similar to the 58% in the overall population. PRP thus seems to enable recovery in a majority of patients who did not respond to hyaluronic acid, and the fact of being naïve to hyaluronic acid or not did not appear to affect the response to PRP. These findings suggest that PRP could be considered as an additional therapeutic option for the management of KOA. Furthermore, our study also shows a reduction in the use of painkillers after PRP. The deleterious effects of certain painkillers are well described, as is the increase in mortality observed in patients treated with Tramadol [41,42]. Moreover, expert recommendations stipulate that PRP can be proposed in as a second-line treatment, after the failure of oral systemic treatment and physiotherapy, but not necessarily after the failure of viscosupplementation. Regarding the treatments taken prior to PRP, we did not observe any significant difference between patients who took prior NSAIDs, antiplatelet, or anticoagulant agents and those who did not, contrary to other studies, such as that of Gupta et al. [43]. However, we deemed that it would put patients at risk of adverse events if they interrupted antiplatelet or anticoagulant therapy, and therefore, patients in our study were allowed to continue these drugs as usual over the course of PRP therapy.

The efficacy of PRP appears to be lower in our study than reported elsewhere. This difference could be explained by several factors. Firstly, we chose to use OMERACT-OARSI response [25] as the primary outcome. It is reliable and validated, but also more stringent. Indeed, if the primary outcome had been a reduction ≥20% in the WOMAC score, then we would have had 64.5% of responders to PRP. This alone could explain the low rate of response observed here. In their study, Alessio-Mazzola et al. [40] considered PRP treatment to have failed if there was symptom recurrence or a need for surgery within 6 months, although no further details were given for this definition. Their PRP protocol also differed from ours, as theirs included four injections. The PRP failure rate observed in their study was only 15.3%, but the differences in study design preclude comparison with our findings, although this explains why we did not identify the same factors of poor prognosis (namely Kellgren–Lawrence grade III and high body mass index in the study by Alessio-Mazzola). Of note, the KOOS-PS [24] is a function score that is simple and quick to use, but it is not widely used in scientific practice. This limits the possibility to interpret and compare results of this score with the literature.

A second possible explanation for the low rate of response in our study is the body mass index and higher initial WOMAC score in our population, compared to other reports. This can be explained by the higher proportion of patients with severe disease (52.9% with Kellgren–Lawrence grade III or IV), although the improvement in WOMAC from before to after treatment was of the same magnitude as reported elsewhere. Despite a higher initial WOMAC score, in the most severely affected patients, we showed a statistically and clinically significant improvement. Efficacy was also observed in patients with Kellgren–Lawrence grade IV. These findings should thus encourage the use of PRP as a valid therapeutic alternative in the management of KOA, regardless of the severity of disease.

Finally, in our study, at the time of the second injection (performed 1 month after the first), 40% of patients had joint effusion, which was drained during the same procedure as the second injection. This rather high rate of joint effusion could partially contribute to the lower efficacy observed here. According the recommendations of the French-speaking expert consensus group, PRP treatment should not be administered during flares of OA, and joint effusion should be systematically drained before PRP injection [18].

In our study, we evaluated the patients’ reported conviction of the efficacy of PRP (as assessed by a score ≥ 7/10), before treatment initiation, to investigate whether patient beliefs and personal convictions may have influenced clinical response. We observed no statistically significant relation, although there was a tendency towards better response among patients who were more convinced of the efficacy of PRP. This may be at least partially explained by the placebo effect. This effect is well-known in interventional rheumatology, where it has been reported that the placebo effect may be large with injectable treatments, reaching up to 21.97 points (95%CI 16.48–27.46) of absolute improvement on the WOMAC pain scale, on a scale of 0 to 100 [44]. However, since the efficacy of PRP is greater than that of hyaluronic acid, the placebo effect alone is unlikely to account for the positive effect of PRP.

## 5. Conclusions

Our study identified a heel-to-buttock distance > 35 cm and ongoing physiotherapy as factors associated with poorer response to treatment among patients with KOA receiving PRP injections. Kellgren stage and BMI were not statistically negatively associated with response. PRP appears to be an effective and well-tolerated therapeutic option after 7 months’ follow-up. These findings confirm the attractiveness of PRP as an additional tool in the therapeutic arsenal for the management of symptomatic KOA after the failure of first-line pharmacological and non-pharmacological treatments.

## Figures and Tables

**Figure 1 diagnostics-13-00760-f001:**
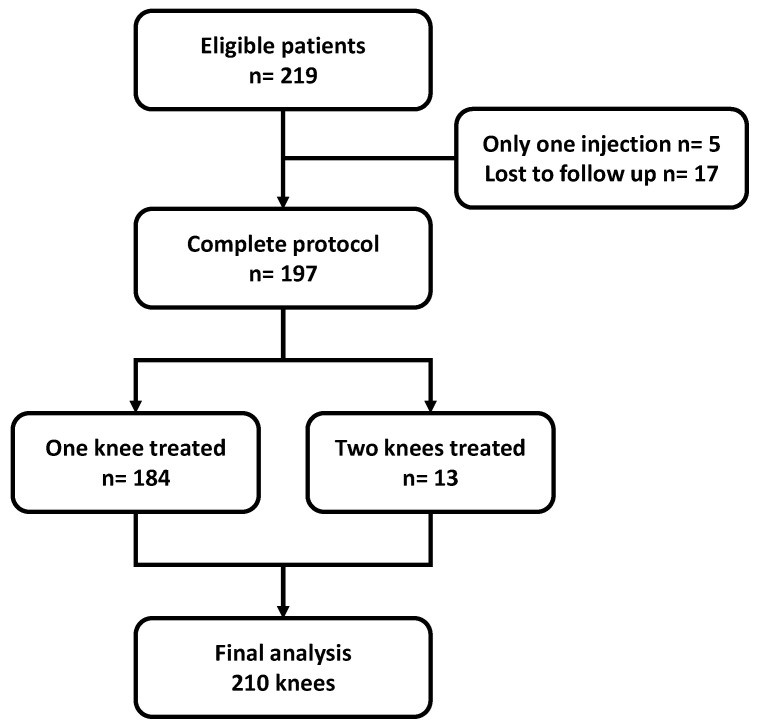
Flowchart of the study. *n* = number of patients.

**Table 1 diagnostics-13-00760-t001:** Characteristics of the study population.

Population	
Number of Patients	197
Number of knees treated	210
Women	135 (64.3)
Mean age, years	59.8 ± 10.8
Mean BMI, kg/m^2^	31.0 ± 6.5
Time since symptom onset, months	36 (12; 72)
Initial WOMAC total score/96	49.4 ± 16.1
Initial VAS score/100	58.5 ± 19.7
Initial KOOS-PS score/28	16.8 ± 4.8
Treatment and comorbidities	
Diabetes mellitus	29 (13.9)
Hypertension	87 (41.6)
Current smokers	28 (11.9)
Dyslipidemia	46 (22.0)
Metabolic syndrome	49 (23.7)
Chronic inflammatory rheumatic disease	38 (18.8)
Prior surgery on treated knee (meniscus/ligaments)	25 (12.0)
Charlson comorbidity index	2.2 ± 1.7
Number of patients with Charlson ≥ 3	85 (40.7)
Aspirin	27 (12.9)
Direct oral anticoagulant	6 (2.9)
Failed viscosupplementation	121 (57.6)
Localization of arthritis	
Internal femorotibial	145 (70.1)
External femorotibial	34 (16.4)
Patellofemoral	71 (34.3)
Tricompartmental	28 (13.5)
Heredity	37 (17.8)
Secondary to chronic inflammatory rheumatic disease	13 (19.7)
Trauma	68 (32.9)
Metabolic	65 (31.3)
Static disorders	100 (48.3)
Kellgren–Lawrence Classification	
- Grade 1	24 (11.8)
- Grade 2	68 (33.5)
- Grade 3	82 (40.4)
- Grade 4	29 (14.3)

**Table 2 diagnostics-13-00760-t002:** Variations over baseline in pain and function scores at 1, 4, and 7 months after PRP injection.

	M0	M1	M4	M7
WOMAC (0–96)	49.2 ± 16.0	43.6 ± 17.9 *	31.1 ± 19.0 *#	31.9 ± 19.8 *#
KOOS-PS (0–28)	16.8 ± 4.8	14.6 ± 5.3 *	12.2 ± 5.9 *#	12.2 ± 6.2 *#
VAS (0–100)	58.5 ± 19.7	51.6 ± 21.7 *	43.7 ± 23.2 *#	43.5 ± 24.6 *#
PASS VAS < 40, *n*(%)	21 (10.0)	41 (22.9) *	76 (39.2) *#	81 (40.9) *#
PASS WOMAC ≤ 46, *n*(%)	80 (38.1)	89 (50.6) *	146 (76.0) *#	148 (75.1) *#

* *p*-value vs. baseline < 0.0001, # *p*-value vs. M1 < 0.0001.

**Table 3 diagnostics-13-00760-t003:** Univariate comparison of responders vs. non-responders.

Characteristics	Responders *n* = 92	Non-Responders *n* = 118	*p*-Value
Age (years)	59 ± 10	60 ± 11	0.41
Females	60 (65.2)	75 (63.6)	0.80
Body mass index, kg/m^2^	30.7 ± 6.9	30.9 ± 6.5	0.87
Platelet count (G/L)	260 ± 54	258 ± 67	0.69
Time since onset of symptoms, months	51 ± 47	72 ± 91	0.27
VAS M0 (0–100)	61.1 ± 18.5	56.5 ± 20.5	0.17
KOOS-PS M0 (/28)	16.4 ± 4.5	17.1 ± 5.0	0.16
Womac M0 (/96)	48.9 ± 14.1	48.9 ± 17.4	0.47
Womac function (/68)	32.4 ± 10.1	34.2 ± 13.3	0.19
Heel-to-buttock distance (cm)	24 ± 10	30 ± 12	0.01
Physiotherapy ongoing	21 (23.3)	46 (40.7)	0.01
Joint effusion	15 (16.3)	20 (17.1)	0.88
Current smokers	12 (13.0)	13 (11.0)	0.65
Metabolic syndrome	17 (18.9)	32 (27.4)	0.16
Chronic inflammatory rheumatic disease	16 (17.4)	22 (18.8)	0.79
Prior surgery of treated knee	9 (9.8)	16 (13.7)	0.39
Chondrocalcinosis	7 (7.78)	11 (9.5)	0.67
Charlson comorbidity index	2.1 ± 1.6	2.3 ± 1.8	0.40
Aspirin	10 (10.9)	17 (14.5)	0.43
Direct oral anticoagulants	2 (2.20)	4 (3.4)	0.70
NSAID use <10 days	9 (10.0)	13 (11.4)	0.75
Failed viscosupplementation	52 (56.5)	69 (58.5)	0.78
Heredity	39 (42.4)	56 (48.7)	0.37
Trauma	29 (31.5)	39 (33.9)	0.72
Secondary to chronic inflammatory rheumatic disease	16 (17.4)	21 (18.1)	0.89
Metabolic	26 (28.3)	39 (33.6)	0.41
Static or biomechanical disorders	45 (49.5)	55 (47.4)	0.77
Kellgren–Lawrence Grade I	13 (14)	11 (10)	0.30
Kellgren–Lawrence Grade II	28 (31)	40 (35)	0.52
Kellgren–Lawrence Grade III	39 (43)	43 (38)	0.84
Kellgren–Lawrence Grade IV	10 (11)	19 (17)	0.25
Internal femorotibial localization	66 (72.5)	79 (68.1)	0.49
External femorotibial localization	16 (17.6)	18 (15.5)	0.69
Patellofemoral	26 (28.6)	45 (38.8)	0.12
Tricompartmental	11 (12.1)	17 (14.7)	0.59

NSAID, non-steroidal anti-inflammatory drugs.

**Table 4 diagnostics-13-00760-t004:** Factors associated with response to PRP by multivariable analysis.

Factor	OR (95%CI)
WOMAC	0.99 (0.97–1.02)
Physiotherapy	0.46 (0.24–0.90)
Patellofemoral localization	0.62 (0.32–1.19)
VAS score ≤ 6	0.55 (0.29–1.05)
Heel-to-buttock distance > 35 cm	0.31 (0.15–0.65)

OR, odds ratio; CI, confidence interval.

## Data Availability

All the data from this study are presented here.

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
