# Peer review of "Prognostic Factors Related to Clinical Response in 210 Knees Treated by Platelet-Rich Plasma for Osteoarthritis"

_diagnostics, 2023, doi:10.3390/diagnostics13040760_

Round 1
Reviewer 1 Report
Dear Authors,
Thank you for addressing the comments raised by me in your prior submission. I only have a minor comment which I suggest should be incorporated:
1. Per my previous comment, you did included a section on preparation of PRP. You also added the platelet concentration compared to whole blood. I would suggest to also classify the PRP type (LR vs LP) - see article link (https://www.mdpi.com/2227-9059/11/1/141); and include the platelet count (as described in this article - https://www.mdpi.com/2227-9059/10/10/2527). Please cite these articles as well.
This is essential to bring more clarity to type of PRP used owing to so much discrepancy about efficacy of PRP in available literature.
Author Response
Thank you for your review. Your previous comments were relevant, and we have taken them into account.
We have added more detail to the section on PRP preparation as you suggested:
“Platelet-rich plasma (PRP) formulations were prepared using the Arthrex ACP ® system. A nurse drew a blood sample of 15 mL, then the sample was centrifuged accord-ing to the manufacturer’s instructions (5 minutes at 1500 rpm), to obtain 3 to 6 mL of PRP, ready for use with the double-syringe system [27]. According to Sundman et al [28], this PRP had low white blood cell content (x0.13 compared to whole blood) and platelets were concentrated x1.99 [28'] to obtain a pure PRP [28'']. Compared with controls (white blood cell count, 8.73 × 109/L), Arthrex ACP® reduces the white blood cell count by almost eliminating the white blood cells to 1.3 × 109/L. The ACP® kit had negligible lympho-cytes (0.7 × 109/L) and neutrophil count (0.4 × 109/L)[28']. Therefore, the PRP used is a LP-PRP which is recommended for KOA according to Milants et al [29]. The PRP is then administered as an IA injection under echographic guidance at M0. A second injection is performed at M1 after preparation of PRP according to the same protocol. The choice to perform one round of centrifugation and to administer 2 injections at 1 month interval was made based on the data described by Milants et al [29]. If joint effusion occurred, nee-dle aspiration was performed at the same time as PRP injection.”
Reviewer 2 Report
i think there is no place for non randomized placebo controlled studies in these type of studies in esteemed journals
Author Response
We believe that this work, although not randomized against placebo, brings elements of answers to daily pratics, which do not have clear answers in the literature to date.
Reviewer 3 Report
Dear Author,
Interesting study, well explained methodology and well written English.
Kind regards,
Dragana Lazarevic
Author Response
Many thanks for your positive appreciation of our work.
Round 2
Reviewer 2 Report
this study does not add to the current knowledge